# On the Statistical Limits of Self-Improving Agents

## Abstract

We develop a learning-theoretic framework for analyzing self-improving agents by decomposing self-modification into five axes. Within this framework, we prove a sharp boundary: under standard i.i.d. assumptions, distribution-free PAC learnability is preserved if and only if the policy-reachable family remains uniformly capacity-bounded. If reachable capacity can grow without bound, utility-rational self-changes can make learnable tasks unlearnable. We further introduce a simple Two-Gate guardrail—a validation-improvement requirement plus a capacity cap—that preserves this boundary and yields standard VC-rate guarantees. The broader implication is that self-modification must be constrained not only by objectives, but also by structural conditions that preserve the statistical prerequisites for learning. As AI systems become increasingly intelligent and autonomous, this framework provides a foundation for the statistical theory of self-improvement.

## 1 Introduction

Current agentic systems increasingly revise parts of their own learning process in response to finite feedback. They may update prompts, code, tools, retrieval memories, model choices, hyperparameters, or update rules across repeated trials. A coding agent that edits a program until visible tests pass, an AutoML loop that retries model and hyperparameter choices on a validation split, and an open-ended self-improvement pipeline that iterates benchmark-driven code edits all share the same statistical pattern: decisions about the learner are themselves being selected on limited evidence.

Classical learning theory—from realizable and agnostic PAC to information-theoretic and computational analyses—typically fixes the learning mechanism in advance. Parameters may adapt, but the update rule family, representation, architecture, computational substrate, and higher-order selection rule are treated as ex ante specified. Self-modification changes this object of analysis. The learner is no longer merely choosing a predictor inside a fixed family; it may also choose which family, update dynamics, or approval rule becomes available next.

The algorithmic axis should therefore be understood as a structured policy-selection problem within standard online and continual learning, not as a new replacement for those settings. Starting from online learners with fixed update rules, we consider learners that may switch among candidate update rules or hyperparameter schedules across time while still defining a single history-dependent learning procedure. When those switches are chosen by myopically maximizing utility on finite evidence, the resulting procedure remains an online or continual learner, but with an additional selection layer that can itself overfit.

This observation extends across the other axes. Representational and architectural edits change what functions can be expressed; substrate edits can change what can be computed or stored; metacognitive edits change which candidate edits are even evaluated. The common question is: when does finite-evidence self-improvement preserve the statistical conditions required for learning, and when can it move the learner into a family too large for the available data?

This question is already visible in contemporary practice. Reinforcement learning and meta-learning instantiate constrained self-change (Sutton & Barto, 2018; Finn et al., 2017; Rajeswaran et al., 2019; Hospedales et al., 2022); automated machine learning and hyperparameter search repeatedly select among model families, schedules, and validation outcomes (Hutter et al., 2019; Feurer & Hutter, 2019; Li et al., 2017); and open-

ended pipelines iterate code edits and tool use (Zhang et al., 2025). Decision-theoretic proposals study provably utility-improving modifications (Schmidhuber, 2005), safety analyses document pathologies (Orseau & Ring, 2011), and metagoal frameworks aim to stabilize goal evolution (Goertzel, 2024). What remains underdeveloped is a learning-theoretic account of post-modification behavior: when do seemingly rational self-changes preserve the conditions under which learning is possible, and when do they destroy them?

We isolate a minimal learning-theoretic invariant governing when self-modification preserves distribution-free generalization guarantees. We prove a sharp policy-level boundary: distribution-free PAC guarantees are preserved if and only if the policy-reachable family has uniformly bounded capacity. We also give a simple Two-Gate guardrail—validation margin $\tau$ together with a capacity cap $K(m)$—that keeps trajectories on the safe side and yields a VC-rate oracle inequality. The broader message is structural: once different forms of self-modification are expressed through the family of predictors they can reach, the learnability question reduces to whether that reachable family remains capacity-bounded.

**Our Contributions.**

- **Sharp learnability boundary.** Under standard i.i.d. assumptions, distribution-free PAC learnability is preserved under self-modification if and only if the policy-reachable family remains uniformly capacity-bounded.

- **Axis decomposition via reachable families.** We show how different forms of self-modification can be analyzed through the reachable family they induce, clarifying which axes expand the effective hypothesis class and which remain structured subsets within standard learning settings.

- **Two-Gate guardrail.** We give a simple computable rule—validation improvement by margin $\tau$ together with a capacity cap $K(m)$—that preserves learnability, guarantees monotone true-risk improvement, and yields a standard VC-rate oracle inequality for the terminal predictor.

**Technical novelty.** The technical novelty is the *policy-reachable family*: the effective hypothesis family induced by the agent's own self-modification policy. This object is neither the initial hypothesis class, nor the largest class imaginable, nor a single post-edit model. It is the family of predictors reachable under the agent's proof-triggered modification rule, including choices made through validation-driven model selection, hyperparameter schedules, code edits, architecture changes, or metacognitive filters. This lets us separate edits that merely change search dynamics from edits that expand the effective family, prove an if-and-only-if learnability boundary at the policy level, and express a concrete guardrail as a validation gate plus a capacity gate. In this sense, the framework applies classical learning theory to a new object induced by self-modification rather than simply restating the fixed-class PAC theorem.

**Scope of the claims.** All main guarantees are stated under standard distribution-free PAC assumptions: i.i.d. data from a fixed distribution, bounded loss, independent validation data where used, and one-axis-at-a-time analysis unless a joint reachable family is explicitly defined. Within this baseline, the framework identifies the invariant any stronger treatment of realistic self-modification must preserve or replace: capacity control of the policy-reachable family.

## 2 Related Work

### 2.1 Decision-theoretic self-modification and safety

Gödel Machines give a proof-based framework for globally optimal self-modification under a utility function (Schmidhuber, 2005). Safety analyses document pathologies for self-modifying agents, including reward hacking and self-termination (Orseau & Ring, 2011). Open-ended empirical systems iterate code and toolchain edits with benchmark gains but without proof obligations (Zhang et al., 2025). Proposals for metagoals aim to stabilize or moderate goal evolution during self-change (Goertzel, 2024). Recent conceptual work argues that truly self-improving agents may require intrinsic metacognitive learning rather than only outer-loop optimization (Liu & van der Schaar, 2025). These frameworks establish decision-theoretic or conceptual

foundations, while leaving open the learning-theoretic question of post-modification generalization. We study that question directly.

## 2.2 Modern mechanisms for self-improvement

Contemporary machine learning already contains constrained forms of self-modification. Neural architecture search explores architectural topologies through differentiable, evolutionary, and reinforcement approaches (Liu et al., 2019; Elsken et al., 2019; Zoph & Le, 2017; Real et al., 2019). Automated machine learning systems perform pipeline, model-family, and hyperparameter search, often through repeated evaluation on held-out data (Hutter et al., 2019; Feurer & Hutter, 2019; Li et al., 2017). Population-based training simultaneously evolves hyperparameters and weights across a population of models (Jaderberg et al., 2017).

Hyperparameter tuning is especially close to our setting. It is sequential, validation-driven, and can either leave capacity essentially fixed, as with a learning-rate schedule, or change the effective family, as with depth, width, kernel features, regularization strength, number of experts, or retrieval size. Standard validation-based model-selection intuitions apply when the candidate family is fixed in advance and capacity is controlled. Our contribution is to express the same issue for agents that can keep modifying the set of candidates itself: a validation gain is not enough unless the policy-reachable family remains bounded relative to the available data.

Meta-learning adapts optimizers, initializations, and inductive biases across tasks (Finn et al., 2017; Rajeswaran et al., 2019; Hospedales et al., 2022). Reinforcement learning and multi-armed bandits provide policies for selecting modifications and exploration strategies (Sutton & Barto, 2018; Auer et al., 2002; Lai & Robbins, 1985; Slivkins, 2019). Representation growth through mixture of experts and adapters, and the use of external memory and retrieval, expand the effective function family and computation available at inference (Fedus et al., 2022; Houlsby et al., 2019; Hu et al., 2021; Graves et al., 2014; 2016; Lewis et al., 2020; Schick et al., 2023). Continual learning addresses sequential task acquisition while mitigating catastrophic forgetting (Kirkpatrick et al., 2017; Parisi et al., 2019; Van de Ven & Tolias, 2022).

These mechanisms instantiate partial self-modification while keeping some surrounding framework fixed. In our framework they traverse representational, architectural, algorithmic, substrate, and metacognitive axes. The point of analysis is not merely that they change over time, but whether the self-improvement policy driving those changes keeps the reachable family statistically safe.

## 2.3 Learning theory for adaptive systems

PAC learning provides distribution-free guarantees under a fixed hypothesis class and algorithm (Shalev-Shwartz & Ben-David, 2014; Mohri et al., 2018; Blumer et al., 1989; Vapnik, 1998; Hanneke et al., 2024). Online learning theory establishes regret bounds for adaptive algorithms (Shalev-Shwartz, 2012; Hazan, 2016; Cesa-Bianchi & Lugosi, 2006), while continual learning studies sequential adaptation under evolving experience (Kirkpatrick et al., 2017; Parisi et al., 2019; Van de Ven & Tolias, 2022); both typically assume that the overall learning mechanism remains fixed even as parameters or task states change. Transformation-invariant learners extend instance equivalence while keeping the learning mechanism fixed (Shao et al., 2022). Predictive PAC relaxes data assumptions with a fixed learner (Pestov, 2010), and iterative improvement within constrained design spaces admits PAC-style analysis (Attias et al., 2025).

Information-theoretic approaches bound generalization for adaptive and meta-learners via mutual information (Jose & Simeone, 2021; Chen et al., 2021; Wen et al., 2025). Stability connects optimization choices to generalization bounds (Bousquet & Elisseeff, 2002; Hardt et al., 2016). These results typically keep some structural object fixed ex ante—for example the hypothesis class, update rule family, architecture, or computational model. We make that structural object explicit under self-modification and characterize when the induced reachable family preserves PAC learnability.

## 2.4 Computability and the substrate

Church–Turing equivalent substrates preserve solvability up to simulation overhead, whereas strictly weaker substrates with finite memory can forfeit learnability of classes that are otherwise PAC-learnable. Stronger-than-Turing models change the problem class under discussion (Akbari & Harrison-Trainor, 2024). This motivates treating substrate edits separately from architectural or representational changes and clarifies when invariance should be expected.

# 3 Setup and Five-Axis Decomposition

**Data and splits.** We study supervised learning under an unknown distribution $\mathcal{D}$ over examples $\langle x, y \rangle \in \mathcal{X} \times \mathcal{Y}$. Let $S \sim \mathcal{D}^m$ denote the training sample and $V \sim \mathcal{D}^{n_v}$ an independent validation sample.

**Modification axes and learner state.** We use the following five components throughout:

$$A = \text{algorithmic}, \quad H = \text{representational}, \quad Z = \text{architectural}, \quad F = \text{substrate}, \quad M = \text{metacognitive}.$$

At time $t \in \mathbb{N}$ the learner state is

$$s_t = \langle A_t, H_t, Z_t, F_t, M_t \rangle \in \underbrace{\mathcal{L}}_{\mathcal{A} \times \mathcal{H} \times \mathcal{Z} \times \mathcal{F} \times \mathcal{M}}.$$

Here $A_t$ denotes update rules or schedules, $H_t$ the effective hypothesis family or representation, $Z_t$ the topology or information-flow structure, $F_t$ the computational substrate and memory semantics, and $M_t$ the scheduler or filter that selects and approves edits.

**Finite evidence and environment context.** Let $\mathcal{E}$ denote the space of finite evidence objects available to the agent, such as a current minibatch, a fixed buffer of past data, a predeclared split of $S$, a validation summary, or other finite statistics derived from interaction history. At time $t$ the agent has finite evidence $E_t \in \mathcal{E}$. We also allow an external environment context $\text{Env}_t$, such as available compute, wall-clock time, or deployment constraints.

**Loss, risk, and utility are distinct objects.** Fix a loss $\ell : \mathcal{Y} \times \mathcal{Y} \to [0,1]$ and define population risk

$$R(h) = \mathbb{E}_{\langle x,y \rangle \sim \mathcal{D}}(\ell(h(x), y))$$

and empirical risks $\widehat{R}_S$ and $\widehat{R}_V$. Utility is a separate agent-internal quantity used to decide whether to self-modify:

$$u(s_t, E_t, \text{Env}_t).$$

It may depend on proxies of risk, validation performance, resources, constraints, or other finite summaries. We do not assume $u$ equals negative loss unless stated.

**Utility-learning tension.** We call the *utility-learning tension* the mismatch between the criterion used to select self-modifications and the conditions required for distribution-free learning. An agent may rationally prefer an edit because it increases its immediate utility $u(s_t, E_t, \text{Env}_t)$ on finite evidence—for example by improving empirical fit, validation performance, efficiency, or other internal objectives—while that same edit expands the policy-reachable family in a way that weakens or destroys the statistical prerequisites for learning, such as uniform capacity control. The central question of this paper is when utility-improving self-modifications remain compatible with learnability, and when they do not.

**Modification map and decision rule.** A possibly stochastic modification map $\Phi : \mathcal{L} \times \mathcal{E} \to \mathcal{L}$ updates the system via

$$s_{t+1} = \Phi(s_t, E_t).$$

For $X \in \{A, H, Z, F, M\}$ with state space $\mathsf{State}_X$, we write

$$X_{t+1} = \Phi_X(X_t, E_t, \theta_{X,t}), \qquad \theta_{X,t} \in \Theta_X,$$

where $\Theta_X$ indexes admissible edits along axis $X$. A candidate modification at time $t$ is executed if and only if there exists a formal proof in the agent's current calculus that it yields an immediate utility increase:

$$u(\Phi(s_t, E_t), E_t, \mathrm{Env}_t) \; > \; u(s_t, E_t, \mathrm{Env}_t).$$

If multiple candidates satisfy this criterion, selection is handled by the metacognitive scheduler $M_t$, such as first-found proof, maximum certified utility gain, or any fixed tie-break rule. Our theorems are stated in terms of the resulting policy-reachable set induced by this selection rule.

**Policy-reachable families.**  For any axis $X \in \{A, H, Z, F, M\}$, let $\mathrm{Reach}_X(u)$ denote the set of $X$-states appearing along some trajectory generated by the decision semantics from $s_0$ under utility $u$.

**Distribution-free PAC learnability and computable PAC.**  A hypothesis family $\mathcal{H} \subseteq \mathcal{Y}^{\mathcal{X}}$ is distribution-free PAC learnable if there exists a learner $\mathsf{Alg}$ such that for all $\varepsilon, \delta \in (0, 1)$, for all distributions $\mathcal{D}$, and for $m \geq m_{\mathcal{H}}[\varepsilon, \delta]$, if $S \sim \mathcal{D}^m$ then with probability at least $1 - \delta$ the output $\hat{h} = \mathsf{Alg}(S)$ satisfies

$$R(\hat{h}) \leq \inf_{h \in \mathcal{H}} R(h) + \varepsilon.$$

When we say learnable in the Turing sense, we additionally require that the learner and the hypotheses it outputs are implementable on a Church–Turing equivalent substrate.

**Capacity notion.**  Our results apply with any uniform capacity notion that yields distribution-free uniform convergence, such as VC-subgraph or pseudodimension. For concreteness, for 0–1 loss we use VC dimension, written $\mathrm{VC}(\cdot)$.

**Constants and notation.**  We use absolute constants $c_1, c_2, \ldots$ whose values may differ across lemmas but are fixed within a statement. We use $\tilde{O}(\cdot)$ to hide polylogarithmic factors in $m, n_v, 1/\delta$. Probabilities are over the draws of $S$ and $V$ unless specified.

**Axis isolation and substrate scope.**  Throughout, we analyze one axis at a time while holding others fixed. Under Church–Turing equivalent substrates $F$, learnability refers to classical PAC in the computable sense. Non Church–Turing cases are treated separately in Section 9.

**Data-path integrity.**  Our PAC statements assume $S$ and $V$ are i.i.d. from $\mathcal{D}$ and independent of each other. Permitted data-path operations are those that preserve i.i.d. draws, such as additional i.i.d. samples, balanced but label-independent subsampling, or predeclared splits. If selection depends on labels or on $V$, standard importance-weighting or covariate-shift corrections must be used; otherwise guarantees may fail.

**Detailed interpretation of the axes.**  Each axis corresponds either to expanding the effective hypothesis family or to filtering trajectories inside a standard model.

1. **Algorithmic.** Update rules, schedules, stopping, internal randomness, and hyperparameter schedules. This axis analyzes a structured subset within standard online and continual learning unless the chosen hyperparameters change the effective family.

2. **Representational.** Changes to the hypothesis class or encoding, such as feature maps, basis expansions, unions, retrieval memories, and refinements. This axis expands the effective hypothesis family.

3. **Architectural.** Topology and information flow, including wiring, routing, depth or width, modular planners, tool routers, and memory addressing. This axis matters through the representational family induced by the architecture.

4. **Substrate.** Computational model and memory semantics, such as the machine model and memory capacity or discipline. This axis affects learnability only via computability changes or by enlarging the induced effective family.

5. **Metacognitive.** A scheduler or filter that selects, orders, and approves modifications. This axis primarily filters trajectories rather than directly enlarging the family.

**Why this decomposition matters.** Self-improvement is often discussed as a monolith, but it is not: agents can change *what* they can represent, *how* they search, *how* information flows, *what* compute they have, and *how* they choose among modifications. The five-axis decomposition makes this explicit and, crucially, makes it analyzable. Each axis induces a set of post-modification predictors, and the union of what the agent can reach under its decision rule is the only object that matters for distribution-free guarantees. Once phrased this way, the safety question becomes concrete: does self-improvement keep the reachable family capacity-bounded, or does it allow capacity to drift without limit? The decomposition therefore turns an amorphous concern—agents that rewrite themselves—into a small number of verifiable invariants.

**Standing assumptions and scope.**

**A1** Data $\langle x, y \rangle$ are i.i.d. from fixed $\mathcal{D}$. Training $S \sim \mathcal{D}^m$ and validation $V \sim \mathcal{D}^{n_v}$ are independent.

**A2** Loss is bounded, $\ell \in [0, 1]$.

**A3** Capacity is any uniform-convergence notion, such as VC, pseudodimension, or VC-subgraph. We instantiate VC where convenient.

**A4** When a computable proxy $\mathcal{B}$ is used, it upper-bounds the chosen capacity notion.

**A5** Substrate semantics. If the substrate $F$ is Church–Turing equivalent, solvability and learnability are measured in the classical computable sense. Non Church–Turing substrates may alter this and are treated separately in Section 9.

**A6** Axis isolation. We analyze one axis at a time while holding the others fixed. Multi-axis edits are discussed later.

**A7** Compute scope. We study sample complexity, not runtime, unless limits are intrinsic to $F$.

## 4   A Simple Multi-Axis Example

Before turning to the theorems, we give one concrete example that combines Axis 1 and Axis 2. Let $\mathcal{X} = [-1, 1]$ and $\mathcal{Y} = \{-1, +1\}$. At time $t$, the learner uses polynomial features up to degree $d_t$, so its representational class is

$$H_t = \left\{ x \mapsto \text{sign}\left(\sum_{j=0}^{d_t} w_j x^j\right) : w \in \mathbb{R}^{d_t+1} \right\}.$$

The learner also chooses an update rule $A_t$ from a finite menu $\mathcal{A}^{\text{cand}}$, for example gradient descent, normalized gradient descent, or a second-order step. Using finite evidence $E_t$, it first chooses

$$A_{t+1} \in \arg \max_{A \in \mathcal{A}^{\text{cand}}} u(\langle A, H_t, Z_t, F_t, M_t \rangle, E_t, \text{Env}_t),$$

and may also enlarge the representation by setting $d_{t+1} = d_t + 1$ whenever

$$u(\langle A_{t+1}, H_{t+1}, Z_t, F_t, M_t \rangle, E_t, \text{Env}_t) > u(\langle A_{t+1}, H_t, Z_t, F_t, M_t \rangle, E_t, \text{Env}_t).$$

This example cleanly separates the roles of the axes. The optimizer switch remains inside standard online or continual learning: it changes *how* the learner traverses a fixed class at that step. The degree increase changes *what* functions are representable, since

$$H_t \subsetneq H_{t+1}.$$

If the degree sequence is globally capped, for example $d_t \leq d_{\max}(m)$, then the reachable family remains capacity-controlled. If myopic utility keeps rewarding degree growth with no such cap, the reachable VC dimension can diverge even though each local edit appeared beneficial.

**Modern tool-using agent interpretation.** The same structure appears in a contemporary LLM-based coding agent. The agent may switch from single-pass generation to a generate-test-repair loop, add retrieval memories or code templates, introduce a planner or verifier, change the execution sandbox, or revise the rule that decides which edits are worth trying. Empirically, this can look like overfitting the agent's sequence of decisions to a visible test suite or benchmark: each edit appears locally justified, but the process can enlarge the space of reachable programs faster than the evidence can support. The polynomial example is an abstraction of the same structural question: what family of predictors or policies becomes reachable after modification? If the reachable family is controlled by a valid cap or proxy, distribution-free guarantees can be retained within the abstraction; if the family can expand without bound, validation gains on finite evidence alone do not imply learnability.

## 5    Representational Self-Modification $\mathcal{M}_H$

**Role of this axis.**    This axis expands the effective hypothesis family.

**Setting.**    We analyze representational edits while holding the algorithmic procedure $A$, architecture $Z$, substrate $F$, and metacognitive rule $M$ fixed. At time $t$ the learner has representation $H_t$ and a representational edit is
$$H_{t+1} = \Phi_H(H_t, E_t, \theta_t).$$
The data, loss, risk, and capacity notation are as defined in Section 3. Proofs are deferred to Appendix D.

**Utility assumptions.**    We distinguish two utility classes and keep them separate throughout the paper. A utility $u$ is *fit-monotone* if it is computable from finite state and evidence, normalized to $[0, 1]$, and non-decreasing in empirical fit on the active finite evidence. For the destruction results only, we use the stronger notion of a *capacity-bonus fit-monotone utility*: in addition to the previous properties, $u$ strictly increases with a computable bonus term $g(\mathrm{VC}(H))$ with $g'(k) > 0$.

**Reference family.**    We work with a fixed capped reference family $\mathcal{G}_{K(m)} \subseteq \mathcal{Y}^{\mathcal{X}}$, satisfying $\mathrm{VC}(\mathcal{G}_{K(m)}) \leq K(m)$ and fixed ex ante before seeing $V$.

**Policy-reachable family.**    For fixed $u$,
$$\mathcal{H}_{\mathrm{reach}}(u) = \left\{ H' : \exists t \text{ along a policy-reachable trajectory under } u \text{ with } H_t = H' \right\}.$$

**Unbounded representational power.**    The pair $\langle \mathcal{H}, \Phi_H \rangle$ has unbounded representational power if for every $m \in \mathbb{N}$ there exist $H, \theta, E$ with
$$\mathrm{VC}(\Phi_H(H, E, \theta)) \geq m.$$
Local unbounded representational power holds if for every $H$ there exists an edit that increases VC by at least one and can fit the current finite evidence.

**Theorem 1. Policy-level learnability boundary**    Under Assumptions A1 through A7, distribution-free learnability is preserved under representational self-modification if and only if there exists $K < \infty$ such that
$$\sup_{H' \in \mathcal{H}_{\mathrm{reach}}(u)} \mathrm{VC}(H') \leq K.$$

*Sketch.* Sufficiency follows because a uniform capacity cap gives uniform convergence on a fixed capped reference family for all steps, and empirical risk minimization (ERM) or approximate empirical risk minimization (AERM) yields the standard VC rate for the terminal predictor. Necessity follows because if capacities along a reachable subsequence diverge, VC lower bounds preclude any distribution-free sample complexity. Full proof appears in Appendix D.

**Theorem 2. Two-Gate finite-sample safety** Given $S$ with $|S| = m$ and independent $V$ with $|V| = n_v$, let $h_{\text{old}} := h_S(H_{\text{old}})$, $h_{\text{new}} := h_S(H_{\text{new}})$, and let $h_T$ denote the terminal accepted predictor. A candidate edit producing $H_{\text{new}}$ is accepted only if

$$\text{Validation} \quad \widehat{R}_V(h_{\text{new}}) \leq \widehat{R}_V(h_{\text{old}}) - [2\varepsilon_V + \tau],$$
$$\text{Capacity} \quad H_{\text{new}} \subseteq \mathcal{G}_{K(m)} \quad \text{and} \quad \text{VC}(\mathcal{G}_{K(m)}) \leq K(m),$$

where $\varepsilon_V$ is chosen so that, with probability at least $1 - \delta_V$ over $V$,

$$\sup_{h \in \mathcal{G}_{K(m)}} \left| R(h) - \widehat{R}_V(h) \right| \leq \varepsilon_V, \qquad \varepsilon_V \asymp \sqrt{\frac{K(m) + \log(1/\delta_V)}{n_v}}.$$

Then with probability at least $1 - \delta_V - \delta$ over draws of $V$ and $S$: each accepted edit decreases true risk by at least $\tau$, and

$$R(h_T) \leq \inf_{h \in \mathcal{G}_{K(m)}} R(h) + \tilde{O}\left(\sqrt{\frac{K(m) + \log(1/\delta)}{m}}\right).$$

**Validation reuse, fixed ex ante.** The same validation set $V$ may be reused adaptively across many edits provided the capped reference family $\mathcal{G}_{K(m)}$ is fixed before seeing $V$ and the gate thresholds $K(m)$, $\varepsilon_V$, and $\tau$ do not depend on $V$. If any of these are tuned using $V$, a fresh split or a reusable holdout is required, as detailed in Appendix D.

**Probability bookkeeping.** All oracle inequalities are stated on the intersection of two events: the uniform validation event on $\mathcal{G}_{K(m)}$ with probability at least $1 - \delta_V$ and the training-side uniform convergence event with probability at least $1 - \delta$. By a union bound, the final probability is at least $1 - \delta_V - \delta$ and does not depend on the number of accepted edits, since the bound is uniform over the fixed capped family.

**Remark.** Under unbounded representational power, utilities that reward empirical fit and even a slight increase in capacity can drive VC unbounded and destroy distribution-free learnability. See Appendix D.

# 6 Architectural Self-Modification $\mathcal{M}_Z$

**Role of this axis.** The architectural axis is not meant to be specific to deep neural networks. It covers any change to topology or information flow—for example a computation graph, planner-verifier decomposition, routing rule, modular tool interface, memory layout, depth, width, or expert mixture. From a PAC perspective, architecture matters through the hypothesis family it induces. Thus the architectural axis can be folded into the representational axis; we keep it separate only because, in modern systems, architecture is often the operational handle by which the representational family is changed.

**Setting and reduction.** We analyze architectural edits while holding the learning algorithm $A$, substrate $F$, and metacognitive rule $M$ fixed. An architecture $Z \in \mathcal{Z}$ induces a hypothesis class $H(Z) \subseteq \mathcal{Y}^{\mathcal{X}}$. At time $t$, an architectural edit produces

$$Z_{t+1} = \Phi_Z(Z_t, E_t, \vartheta_t), \qquad h_S(Z_{t+1}) \in \arg\min_{h \in H(Z_{t+1})} \widehat{R}_S(h).$$

Fix a fit-monotone utility $u$. Let the policy-reachable architectures and induced classes be

$$\mathcal{Z}_{\text{reach}}(u) = \left\{ Z' : \exists t \text{ on a proof-triggered trajectory from } Z_0 \text{ under } u \text{ with } Z_t = Z' \right\},$$

and

$$\mathcal{H}_{\text{reach}}^Z(u) = \left\{ H(Z) : Z \in \mathcal{Z}_{\text{reach}}(u) \right\}.$$

**Why the reduction is useful.** The reduction to representation is intentional rather than a separate proof burden. It says that architectural self-modification has no independent PAC magic: once an architecture induces a class $H(Z)$, the learnability question is exactly whether the set of classes reachable through architectural edits remains capacity-bounded. This also explains why architectural edits can be analyzed together with representational edits in a joint reachable family when a system changes both at once.

**Utility realism.** The boundary and Two-Gate guarantees depend only on the capacity of the reachable family, not on explicit capacity rewards. Even if $u$ has no bonus term, any policy that permits capacity-increasing edits can cross the boundary unless a cap such as $K(m)$ is enforced. The stronger capacity-bonus variant is used only for destruction results.

Every run of $\mathcal{M}_Z$ corresponds to a run of $\mathcal{M}_H$ over the induced family $\mathcal{H}^Z_{\mathrm{reach}}(u)$.

**Lemma 3. Architectural to representational reduction** For any fit-monotone $u$, every proof-triggered trajectory $Z_0 \to Z_1 \to \cdots$ induces a representational trajectory $H(Z_0) \to H(Z_1) \to \cdots$ over a fixed reference family, with ERM or AERM inside each accepted $H(Z_t)$. Consequently $\mathcal{H}^Z_{\mathrm{reach}}(u) = \{H(Z) : Z \in \mathcal{Z}_{\mathrm{reach}}(u)\}$. *Proof sketch.* The decision semantics and utility are unchanged by renaming states from $Z$ to the induced $H(Z)$.

**Theorem 4. Architectural boundary via induced reachable family** For any fit-monotone $u$, distribution-free PAC learnability under architectural self-modification is preserved if and only if

$$\sup_{Z \in \mathcal{Z}_{\mathrm{reach}}(u)} \mathrm{VC}(H(Z)) \ \leq \ K \ < \ \infty.$$

Equivalently, preservation holds if and only if $\sup_{H' \in \mathcal{H}^Z_{\mathrm{reach}}(u)} \mathrm{VC}(H') \leq K$.

*Proof.* Immediate by reduction to Section 5. Apply the representational boundary theorem, Theorem 1, to the induced set $\mathcal{H}^Z_{\mathrm{reach}}(u)$. $\qquad\qquad\square$

**Reference family and proxy-cap subfamily.** Fix a single parameterized super-family $\mathcal{G} \subseteq \mathcal{Y}^{\mathcal{X}}$ that contains every induced class: $H(Z) \subseteq \mathcal{G}$ for all $Z$. For $K \in \mathbb{N}$ define the proxy-cap subfamily

$$\mathcal{G}^{\mathrm{proxy}}_K = \{h \in \mathcal{G} : \ \exists Z \text{ with } \mathcal{B}(Z) \leq K \text{ and } h \in H(Z)\},$$

where $\mathcal{B}(Z)$ is a computable architectural capacity proxy satisfying $\mathrm{VC}(H(Z)) \leq \mathcal{B}(Z)$. Since each accepted $H(Z)$ is a subset of the fixed reference subfamily $\mathcal{G}^{\mathrm{proxy}}_K$, capacity is bounded by $\mathrm{VC}(\mathcal{G}^{\mathrm{proxy}}_K) \leq K$.

**Corollary 5. Two-Gate safety for architecture** Let the capacity gate enforce $H(Z_{\mathrm{new}}) \subseteq \mathcal{G}^{\mathrm{proxy}}_{K(m)}$ with $\mathrm{VC}(\mathcal{G}^{\mathrm{proxy}}_{K(m)}) \leq K(m)$, and the validation gate enforce $\widehat{R}_V(h_S(Z_{\mathrm{new}})) \leq \widehat{R}_V(h_S(Z_{\mathrm{old}})) - [2\varepsilon_V + \tau]$, where $\varepsilon_V$ is chosen by a VC bound on $\mathcal{G}^{\mathrm{proxy}}_{K(m)}$. Then the safety and rate conclusions hold as stated by Theorem 2.

**Validation reuse.** We fix $\mathcal{G}^{\mathrm{proxy}}_{K(m)}$, the schedule $K(m)$, and thresholds ex ante before seeing $V$. If any choice is tuned on $V$, we use a fresh split or a reusable holdout mechanism; all theorems apply to the fixed family.

**Local architectural edits.** We say $\mathcal{M}_Z$ has local unbounded representational power if for every $Z$ there exists a computable edit $\langle \vartheta, E \rangle$ such that $\mathrm{VC}(H(\Phi_Z(Z, E, \vartheta))) \geq \mathrm{VC}(H(Z)) + 1$ and the new class can interpolate the current finite evidence.

**Proposition 6. Robust destruction under local power** Assume local unbounded representational power for $\mathcal{M}_Z$. For any capacity-bonus fit-monotone utility $u$, there exist a distribution $\mathcal{D}$ and sample size $m$ such that the proof-trigger repeatedly accepts local architectural edits that increase capacity, the induced reachable set $\mathcal{H}^Z_{\mathrm{reach}}(u)$ has unbounded VC, and distribution-free PAC learnability fails. *Sketch.* Each local step strictly increases $u$, so the proof-trigger fires. Iteration yields unbounded VC, then apply Theorem 4.

**Proposition 7. Proxy-cap sufficiency**  If $\mathcal{B}(Z) \geq \mathrm{VC}(H(Z))$ for all $Z$ and the capacity gate enforces $\mathcal{B}(Z_{\mathrm{new}}) \leq K(m)$, then the Two-Gate guarantee holds with

$$\mathcal{G}_{K(m)} = \{h \in \mathcal{G} : \exists Z \text{ with } \mathcal{B}(Z) \leq K(m) \text{ and } h \in H(Z)\},$$

and with the same VC-rate bound.

## 7 Metacognitive Self-Modification $\mathcal{M}_M$

**Role of this axis.**  This axis primarily filters trajectories rather than directly enlarging the effective hypothesis family.

**Setting.**  We analyze metacognitive scheduling and filters $M$ while holding representation $H$, architecture $Z$, algorithm $A$, and substrate $F$ fixed. The rule $M$ selects which edits to evaluate and applies acceptance or rejection using finite evidence.

**Metacognitive scheduler or filter.**  A metacognitive rule $M$ can choose which candidate edit to evaluate, when to evaluate, and can randomize; it then accepts or rejects using only finite evidence, such as $S$, $V$, a capacity proxy, and edit costs. Let

$$\mathcal{H}_{\mathrm{reach}}^{M,H}(u) = \{H' : \Pr(\exists t \text{ on a proof-triggered trajectory from } H_0 \text{ under } u \text{ filtered by } M \text{ with } H_t = H') > 0\}.$$

Randomized $M$ is allowed. All guarantees hold almost surely over the internal randomness of $M$.

**Theorem 8. Boundary under metacognitive filtering**  For any fit-monotone $u$ and metacognitive filter $M$, distribution-free PAC learnability is preserved if and only if

$$\sup_{H' \in \mathcal{H}_{\mathrm{reach}}^{M,H}(u)} \mathrm{VC}(H') \leq K < \infty.$$

*Proof sketch.* Apply the representational boundary theorem to the $M$-filtered family. Necessity follows by the same destruction argument when the supremum is unbounded. Full details appear in Appendix F.

**Two-Gate as metacognition and gate integrity.**  If $M$ accepts only when the validation margin holds and $\mathrm{VC}(H_{t+1}) \leq K(m)$, then with probability at least $1 - \delta_V - \delta$ each accepted edit reduces true risk by at least $\tau$, and

$$R(h_T) \leq \inf_{h \in \mathcal{G}_{K(m)}} R(h) + \tilde{O}\left(\sqrt{\frac{K(m) + \log(1/\delta)}{m}}\right),$$

by Theorem 2. This guarantee is conditional on *gate integrity*: the metacontroller enforcing Two-Gate must itself remain fixed, or any admissible modification to it must preserve the same capped reachable-family invariant. Otherwise, a utility-rational agent may face pressure to weaken the very constraint that certifies learnability, creating a higher-order form of meta-decision collapse. Thus Two-Gate should be interpreted not as an automatically self-stabilizing policy, but as a guardrail whose own update rule must be protected by immutability, external enforcement, or a recursively applied meta-gate.

**Meta-stability and recursive guarding.**  This reveals a simple regress: if the agent may optimize the guardian, then the guardian must also be guarded. In our framework, the natural resolution is recursive invariant preservation: every admissible edit to $M$ must itself certify that the post-edit reachable family remains uniformly capacity-bounded. We do not claim to fully solve this regress here; rather, we make explicit that metacognitive safety requires not only a gate on object-level edits, but also a protection mechanism for the gate itself.

**Restorative metacognition.**  Suppose the unfiltered reachable family satisfies $\sup_{H' \in \mathcal{H}_{\mathrm{reach}}(u)} \mathrm{VC}(H') = \infty$. There exists a metacognitive rule $M$, such as Two-Gate with any computable nondecreasing schedule $K(m)$ and preserved gate integrity, such that $\sup_{H' \in \mathcal{H}_{\mathrm{reach}}^{M,H}(u)} \mathrm{VC}(H') \leq K < \infty$, hence learnability is preserved.

**Edit efficiency under margins.** Under Two-Gate with margin $\tau > 0$, along any accepted trajectory the number of accepted edits is at most

$$\frac{R(h_0) - R^\star}{\tau}, \qquad R^\star = \inf_{h \in \mathcal{G}_{K(m)}} R(h),$$

where $h_0$ denotes the initial accepted predictor on the trajectory. Each accepted edit decreases true risk by at least $\tau$, so the bound follows by telescoping.

**Scheduling and randomness.** Because the deviation bound is uniform over the capped family, the safety and rate guarantees hold for every realized trajectory filtered by $M$. Scheduling affects efficiency, not the boundary.

## 8 Algorithmic Self-Modification $\mathcal{M}_A$

**Role of this axis.** This axis concerns policy-selected update dynamics within a fixed effective hypothesis family.

**Setting.** We analyze algorithmic self-modification while holding the hypothesis class $H$, architecture $Z$, substrate $F$, and metacognitive rule $M$ fixed. The agent may change its update rule, optimizer, schedule, stopping rule, or capacity-neutral hyperparameters across time, producing

$$A_{t+1} = \Phi_A(A_t, E_t, \vartheta_t).$$

Given a training set $S$ of size $m$, the realized self-modified update schedule $A_{0:T}$ together with $H$ determines the output predictor $\hat{h} = \mathsf{Alg}(A_{0:T}, S, H) \in H$.

**Relation to standard online and continual learning.** We treat optimizer and update-rule switching as a structured subclass of standard online and continual learning procedures. Any procedure that switches optimizers—for example, run SGD then switch to Newton once a condition holds—is still an online learning algorithm because it defines a single map from histories to actions or updates; likewise, under sequential nonstationary experience it is still a continual learner. Our focus is a structured subset of such algorithms: those that choose among a family of candidate update rules using finite evidence $E_t$ to myopically increase utility. This induces an algorithm-selection effect: when the family of reachable update rules has high statistical flexibility relative to the available evidence, the agent can overfit the *choice of update rule* to $E_t$, yielding long-run degradation even when each candidate update rule is itself valid.

**Hyperparameter tuning.** Hyperparameter tuning sits at the boundary between algorithmic and representational self-modification. Tuning a learning rate, batch schedule, optimizer, or stopping rule while keeping $H$ fixed is algorithmic. Tuning depth, width, kernel features, regularization strength, retrieval size, number of experts, or any choice that changes the induced class is representational or architectural. Existing validation-based hyperparameter search is therefore a special case of our setup when the candidate set is fixed ex ante. The additional risk in self-modifying systems is that the agent may modify the candidate set itself, so the validation split certifies improvement only if the resulting reachable family remains capacity-controlled.

**Takeaways.** Algorithmic edits cannot cure infinite capacity. On finite capacity, empirical risk minimization (ERM) or approximate empirical risk minimization (AERM) preserves PAC. When self-modification alters training dynamics beyond ERM assumptions, a simple stability meta-policy based on step-mass controls the generalization gap.

**Proposition 9. No algorithmic cure for infinite VC** If $\mathrm{VC}(H) = \infty$, then no distribution-free PAC guarantee is possible for any algorithmic procedure. In particular, algorithmic self-modification cannot restore distribution-free learnability.

**Proposition 10. Finite VC is sufficient with ERM or AERM**   If $\mathrm{VC}(H) \leq K < \infty$ and the possibly self-modified training procedure is ERM or AERM over $H$, then for any $\delta \in (0,1)$, with probability at least $1 - \delta$ over $S \sim \mathcal{D}^m$,

$$R(\hat{h}) \;\leq\; \inf_{h \in H} R(h) \;+\; \tilde{O}\!\left( \sqrt{\frac{K + \log(1/\delta)}{m}} \right).$$

**Stability meta-policy via step-mass.**   Assume bounded, Lipschitz, and smooth losses as formalized in Appendix G, and that training examples are sampled uniformly from $S$ during updates. Let a self-modified training run on $H$ use SGD-like updates with step sizes $\{\eta_t\}_{t=1}^T$. Define the step-mass $M_T = \sum_{t=1}^T \eta_t$.

**Theorem 11. Algorithmic stability via step-mass**   Under the conditions above, there exists a constant $C > 0$ independent of $m$ such that

$$\mathbb{E}\Big( R(\hat{h}) - \widehat{R}_S(\hat{h}) \Big) \;\leq\; \frac{C}{m} \sum_{t=1}^T \eta_t \;=\; \frac{C}{m} \, M_T.$$

A metacognitive rule that caps $M_T \leq \Lambda(m)$ guarantees $\mathbb{E}[\mathrm{gap}] = \tilde{O}(\Lambda(m)/m)$. Choosing $\Lambda(m) = \tilde{O}(1)$ yields a $\tilde{O}(1/m)$ gap.

**Discussion.**   Proposition 9 says capacity, not optimizer choice, governs distribution-free learnability. Proposition 10 ensures algorithmic edits that continue to output ERM or AERM do not harm PAC guarantees when VC is finite. Theorem 11 offers a simple meta-policy: cap cumulative step-mass to keep the generalization gap small during algorithmic self-modification. Full proofs are in Appendix G.

# 9   Substrate Self-Modification $\mathcal{M}_F$

**Role of this axis.**   This axis affects learnability only through computability or by changing the induced effective family.

**Setting.**   We analyze substrate edits while holding the specification of $H$, $Z$, and $A$ fixed. Switching substrates changes how these are executed but not which hypotheses are definable nor which utilities are expressible. PAC learnability is classical in the computable sense unless noted.

A substrate edit is

$$F_{t+1} = \Phi_F(F_t, E_t, \varphi_t).$$

**Takeaways.**   First, switching among Church–Turing equivalent substrates preserves classical PAC learnability. Second, downgrading to a strictly weaker substrate, such as finite-state memory, can destroy PAC learnability even for problems learnable pre-switch. Third, stronger-than-Church–Turing substrates do not alter classical PAC guarantees unless they enlarge the induced hypothesis family. If they do, the policy-reachable boundary from Section 5 governs the enlarged family.

**Theorem 12.   Church–Turing invariance of PAC learnability**   If $F$ and $F'$ are Church–Turing equivalent, then a problem that is distribution-free PAC learnable when run on $F$ remains distribution-free PAC learnable when run on $F'$, with the same sample complexity up to constant factors. Computation may differ.

**Proposition 13. Finite-state downgrade can destroy learnability**   There exists a binary classification problem that is PAC learnable on a Church–Turing equivalent substrate but becomes not distribution-free PAC learnable after switching to a fixed finite-state substrate with bounded persistent memory, even when $H$ has finite VC dimension as a specification.

**Proposition 14. Beyond Church–Turing substrates**  If a stronger-than-Church–Turing substrate $F^\dagger$ does not enlarge the induced hypothesis family, classical PAC learnability is unchanged. If it enlarges the effective family to $H^\dagger$, the learnability boundary is governed by

$$\sup_{H' \in \mathcal{H}^\dagger_{\mathrm{reach}}(u)} \mathrm{VC}(H'),$$

exactly as in Section 5.

**Discussion.**  Theorem 12 elevates substrate choice out of the classical PAC calculus: Church–Turing equivalent machines affect compute, not sample complexity. Proposition 13 formalizes that collapsing persistent memory imposes an information bottleneck that breaks distribution-free guarantees. Proposition 14 shows that any real change to learnability arises only through the induced hypothesis family. The policy-reachable boundary applies verbatim once that family changes. Full proofs appear in Appendix H.

## 10  Limitations and Scope

**Standard i.i.d. baseline.**  The main theorems are intentionally stated in the classical distribution-free PAC setting. The training and validation samples are independent i.i.d. draws from a fixed distribution, losses are bounded, and the accepted hypotheses lie in a fixed capped reference family before validation is inspected. These assumptions define the baseline in which the central obstruction can be stated sharply: unbounded policy-reachable capacity rules out distribution-free learnability.

**Nonstationary, adaptive, and interactive data.**  If data are collected online, selected adaptively, or drawn from a changing distribution, the same invariant must be paired with concentration tools suited to the data-generating process. In those settings, one would need additional machinery such as martingale concentration, online-to-batch conversion, covariate-shift or importance-weighting assumptions, reusable holdout methods, or explicit mixing and drift conditions. The same conceptual invariant may remain useful, but the validation and training events must be replaced by concentration statements appropriate to the data-generating process.

**Axis isolation and causality.**  The five-axis decomposition is an analytical device for exposing how each axis induces or filters a reachable family. We analyze one axis at a time to make these effects explicit. For realistic agents that modify several axes simultaneously, the relevant object is the jointly induced reachable family, and guarantees require a global cap on that joint family rather than separate informal assurances for each component.

**Capacity proxies.**  Two-Gate is only as reliable as the capacity proxy used in its capacity gate. A computable upper bound $\mathcal{B}(\cdot)$ may be conservative, loose, or difficult to maintain under composition. The framework therefore identifies what must be controlled, but practical systems still require domain-specific proxy design, auditing, and validation.

## 11  Outlook

**From theory to practice, why capacity bounds matter.**  Modern deep learning often succeeds far beyond what worst-case PAC sample-complexity bounds would predict, often because of implicit regularization, favorable inductive biases, and benign data structure. Self-modification changes the role of capacity control. A fixed high-capacity model may still generalize well, but a self-modifying learner can repeatedly accept capacity-increasing edits on finite evidence, eventually entering regimes where no distribution-free guarantee remains available. Capacity control is therefore not a claim that overparameterized models fail; it is the condition under which worst-case statistical guarantees remain meaningful when the learner can change the family it is learning over.

The Two-Gate policy suggests a direct operational interpretation: track a computable capacity proxy $\mathcal{B}(\cdot)$, set a schedule $K(m)$ relative to available data, and reject edits unless validation improves by margin $\tau$. These

checks are lightweight and are intended as a sufficient guardrail within our abstraction. The central point is not that implicit regularization becomes irrelevant, but that once self-modification is allowed, relying on it alone provides no general distribution-free guarantee.

**Multi-axis modification: the realistic frontier.** Real autonomous agents may modify several components at once: architectures, update rules, tool libraries, and metacognitive policies. Our framework suggests a common way to analyze such settings: the relevant object is the jointly induced reachable family. Learnability is preserved only if that family remains capacity-bounded. This implies that capacity control must ultimately be global rather than purely per-axis, since interactions between individually innocuous edits may still produce an unbounded effective family.

A key open problem is therefore practical rather than conceptual: how to construct computable capacity proxies that remain valid under composition. The gap between a tractable upper bound $\mathcal{B}(\cdot)$ and the true capacity of the induced family determines how conservative a Two-Gate policy must be in realistic systems.

**Towards sustainable self-improvement.** The perspective developed here suggests a simple design principle: self-improvement should be evaluated not only by whether an edit improves near-term utility, but also by whether it preserves the statistical conditions required for continued learning. In our setting, this means constraining reachable capacity relative to available data. Such constraints do not preclude improvement; they instead distinguish sustainable self-improvement from self-modification that outruns the evidence needed to justify it.

## 12 Conclusion

We established a sharp learnability boundary for self-modifying agents: distribution-free PAC guarantees are preserved if and only if the policy-reachable hypothesis family has uniformly bounded capacity. This boundary provides a common statistical criterion across representational, architectural, metacognitive, algorithmic, and substrate modifications once each is expressed through the family of predictors reachable under the agent's policy. We further gave a simple Two-Gate guardrail—a validation margin together with a capacity cap—that enforces this boundary and yields standard oracle-style guarantees for the terminal predictor.

Safe self-improvement therefore requires not only favorable objectives but structural constraints on reachable families. We offer this framework deliberately ahead of the systems that will require it: as agents acquire greater autonomy over their own learning mechanisms, knowing where the statistical boundary lies is a prerequisite for staying on the right side of it.

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

# A    Appendix

# B    Definitions

| Symbol | Definition |
|---|---|
| $s_t$ | Learner state at time $t$, namely $s_t = \langle A_t, H_t, Z_t, F_t, M_t \rangle$ |
| $E_t$ | Finite evidence available at time $t$ (e.g., minibatch, buffer, predeclared split, finite summary) |
| $\Phi, \Phi_X$ | Global modification map and its axis-specific component maps |
| $\mathcal{M}_H, \mathcal{M}_Z, \mathcal{M}_M, \mathcal{M}_A, \mathcal{M}_F$ | Representational, architectural, metacognitive, algorithmic, and substrate self-modification axes, respectively |
| $\Theta_X$ | Admissible edit parameters for axis $X$ |
| $\mathrm{Env}_t$ | External environment context (e.g., compute budget, wall-clock time, deployment constraints) |
| $H(Z)$ | Hypothesis class induced by architecture $Z$ |
| $\mathcal{H}_{\mathrm{reach}}(u)$ | Policy-reachable representational family under utility $u$ |
| $\mathcal{H}_{\mathrm{reach}}^Z(u)$ | Architectural reachable family induced by reachable architectures |
| $\mathcal{H}_{\mathrm{reach}}^{M,H}(u)$ | Reachable family under metacognitive filtering |
| $\mathcal{B}(\cdot)$ | Computable capacity proxy (upper-bounds the chosen capacity notion, e.g., VC/pseudodimension) |
| $K(m)$ | Nondecreasing capacity cap schedule at sample size $m$ |
| $\mathcal{G}_{K(m)}$ | Fixed reference subfamily with capacity at most $K(m)$ |
| $\mathcal{G}_K^{\mathrm{proxy}}$ | Proxy-capped architectural reference subfamily |
| $H_{\mathrm{old}}, H_{\mathrm{new}}$ | Current and candidate hypothesis classes in a Two-Gate decision |
| $h_{\mathrm{old}}, h_{\mathrm{new}}$ | Predictors before and after a candidate edit, defined by ERM/AERM within the corresponding class |
| $h_0$ | Initial accepted predictor along a self-modification trajectory |
| $h_T$ | Terminal accepted predictor along a self-modification trajectory |
| $\varepsilon_V, \delta_V$ | Validation deviation radius and its confidence parameter |
| $\tau$ | Validation margin in Two-Gate |
| $\widehat{R}_S, \widehat{R}_V$ | Empirical risks on train $S$ and validation $V$ |
| $A_{0:T}$ | Realized self-modified update schedule from time 0 through $T$ |
| $M_T$ | Step-mass $\sum_{t=1}^T \eta_t$ in algorithmic self-modification |
| $\Lambda(m)$ | Upper bound schedule for allowed step-mass |
| $R^\star$ | $\inf_{h \in \mathcal{G}_{K(m)}} R(h)$ |

Table 1: Notation used throughout the main text and appendix.

# C    Assumption Scope and Use in the Proofs

**Where i.i.d. sampling is used.** The i.i.d. assumption is used only to invoke uniform convergence for the fixed capped reference families appearing in the main text. In the representational and architectural proofs, it gives the training-side VC event for ERM or AERM. In the Two-Gate proof, it also gives the validation-side event controlling $\sup_{h \in \mathcal{G}_{K(m)}} |R(h) - \widehat{R}_V(h)|$.

**Why validation must be independent or protected.** The validation proof assumes that the reference family $\mathcal{G}_{K(m)}$, the cap schedule $K(m)$, and the margin parameters are fixed before inspecting $V$. Adaptive reuse of the same validation set is safe here only because the deviation event is uniform over that fixed family. If the family, cap, thresholds, or edit proposals are tuned using validation outcomes, then a fresh validation split, a reusable holdout, or an explicit adaptive-data analysis is required.

**What changes outside the baseline.** For non-i.i.d., nonstationary, adversarial, or interactively collected data, the PAC statements should be read as baseline guarantees rather than direct deployment guarantees. Extending the results would require replacing the VC uniform-convergence events with the appropriate concentration or regret statements for the data process, such as martingale bounds, online-to-batch conversion, mixing assumptions, shift corrections, or importance weighting. The policy-reachable-family invariant remains the proposed structural object, but the proof technique must change.

**Axis isolation.** The one-axis proofs hold other state components fixed to identify the statistical effect of each modification type. For simultaneous multi-axis edits, the same logic applies only after defining the joint reachable family induced by the combined policy. Separate caps on individual axes are sufficient only when they imply a cap on the joint family.

## D Full Proofs for Representational Self-Modification

**Data, loss, risks.** Samples $(x, y) \sim \mathcal{D}$ i.i.d. Training $S \sim \mathcal{D}^m$ and validation $V \sim \mathcal{D}^{n_v}$ are independent. Loss $\ell \in [0, 1]$; true risk $R(h) = \mathbb{E}_{(x,y) \sim \mathcal{D}}[\ell(h(x), y)]$. Empirical risks are $\widehat{R}_S$ and $\widehat{R}_V$ on $S$ and $V$.

**Representational edits and policies.** At time $t$ the representation is a hypothesis class $H_t \subseteq \mathcal{Y}^{\mathcal{X}}$. A representational edit is

$$H_{t+1} = \Phi_H(H_t, E_t, \theta_t),$$

where $E_t$ is finite evidence and $\theta_t$ are edit parameters. Within any accepted class $H$, the learner outputs an empirical risk minimization (ERM) or approximate empirical risk minimization (AERM) predictor on $S$:

$$h_S(H) \in \arg\min_{h \in H} \widehat{R}_S(h).$$

The decision rule executes an edit only when an immediate utility increase is formally provable from finite evidence.

**Utility classes.** A utility $u$ is *fit-monotone* if it is computable from finite state and evidence and non-decreasing in empirical fit on the active finite evidence. For the destruction results only, we use the stronger notion of a *capacity-bonus fit-monotone utility*: such a utility is fit-monotone and additionally contains a strictly increasing computable bonus $g(\text{VC}(H))$ with $g'(k) > 0$. We normalize $u \in [0, 1]$ without loss of generality.

**Policy-reachable family.** Fix $u$. Let $\mathcal{H}_{\text{reach}}(u)$ be the set of classes $H'$ for which there exists a time $t$ on some proof-triggered trajectory from $H_0$ under $u$ with $H_t = H'$.

**URP and Local-URP.** The pair $(\mathcal{H}, \Phi_H)$ has *URP* if for every $m \in \mathbb{N}$ there exist $(H, \theta, E)$ with

$$\text{VC}(\Phi_H(H, E, \theta)) \geq m.$$

It has *Local-URP* if for every $H$ there is an edit $(\theta, E)$ such that

$$\text{VC}(\Phi_H(H, E, \theta)) \geq \text{VC}(H) + 1$$

and the new class can interpolate the current finite evidence (e.g., fit $S$).

**Single capped reference family (indexing control).** For each $K \in \mathbb{N}$ let $\mathcal{G}_K \subseteq \mathcal{Y}^{\mathcal{X}}$ be a reference family with $\text{VC}(\mathcal{G}_K) \leq K$ and assume the capacity gate guarantees all accepted $H$ satisfy $H \subseteq \mathcal{G}_K$. This avoids pathologies from taking unions over arbitrarily many distinct classes with the same VC cap. The family $\mathcal{G}_K$ is fixed independently of the validation sample; otherwise the uniform validation event must be replaced by an adaptive-validation argument.

**VC uniform convergence.** There exists a universal constant $c_1 > 0$ such that for any class $G$ with $\mathrm{VC}(G) \leq K$ and any $\delta \in (0,1)$, with probability at least $1 - \delta$ over a sample of size $n$,

$$\sup_{h \in G} \big| R(h) - \widehat{R}(h) \big| \ \leq \ c_1 \sqrt{\frac{K + \log(1/\delta)}{n}}. \tag{1}$$

We hide polylogarithmic factors in $\tilde{O}(\cdot)$.

## D.1  Sharp policy-level boundary

**Theorem (Sharp boundary; restated from Thm. 1).** For any fit-monotone $u$, distribution-free PAC learnability is preserved under representational self-modification if and only if there exists $K < \infty$ such that

$$\sup_{H' \in \mathcal{H}_{\mathrm{reach}}(u)} \mathrm{VC}(H') \ \leq \ K.$$

**Proof (sufficiency).** Fix $u$ and assume $\sup_{H' \in \mathcal{H}_{\mathrm{reach}}(u)} \mathrm{VC}(H') \leq K$. Along any policy-reachable run, all classes satisfy $H_t \subseteq \mathcal{G}_K$ with $\mathrm{VC}(\mathcal{G}_K) \leq K$. By (1) applied to $\mathcal{G}_K$ and ERM in $H_t$,

$$R\big(h_S(H_t)\big) \ \leq \ \inf_{h \in H_t} R(h) \ + \ \tilde{O}\Big(\sqrt{\tfrac{K + \log(1/\delta)}{m}}\Big)$$

uniformly for all $t$, with probability at least $1 - \delta$ over $S$. In particular the terminal predictor $h_T$ obeys the same bound, so $m = \tilde{O}\big((K + \log(1/\delta))/\epsilon^2\big)$ suffices for $(\epsilon, \delta)$-accuracy. $\qquad \square$

**Proof (necessity).** If $\sup_{H' \in \mathcal{H}_{\mathrm{reach}}(u)} \mathrm{VC}(H') = \infty$, then for each $k$ there exists a reachable $H^{(k)}$ with $\mathrm{VC}(H^{(k)}) \geq k$. Classical VC lower bounds imply any distribution-free learner needs $m = \Omega(k/\epsilon)$ samples for $(\epsilon, \delta)$-accuracy (even realizable). Since $k$ is unbounded along reachable trajectories, no uniform PAC guarantee exists. $\qquad \square$

## D.2  Finite-sample safety of the Two-Gate policy

**Two gates.** Given train $S$ ($|S| = m$) and independent validation $V$ ($|V| = n_v$), let $h_{\mathrm{old}} := h_S(H_{\mathrm{old}})$, $h_{\mathrm{new}} := h_S(H_{\mathrm{new}})$, and let $h_T$ denote the terminal accepted predictor. Accept an edit only if

$$\textbf{(Validation)} \quad \widehat{R}_V(h_{\mathrm{new}}) \ \leq \ \widehat{R}_V(h_{\mathrm{old}}) \ - \ (2\varepsilon_V + \tau),$$

$$\textbf{(Capacity)} \quad H_{\mathrm{new}} \ \subseteq \ \mathcal{G}_{K(m)} \quad \text{with} \quad \mathrm{VC}(\mathcal{G}_{K(m)}) \leq K(m),$$

where $K(\cdot)$ is nondecreasing, $\tau \geq 0$ is a margin, and $\varepsilon_V$ is chosen so that with probability at least $1 - \delta_V$ over $V$,

$$\sup_{h \in \mathcal{G}_{K(m)}} \big| R(h) - \widehat{R}_V(h) \big| \ \leq \ \varepsilon_V \quad (\text{e.g., } \varepsilon_V \asymp \sqrt{(K(m) + \log(1/\delta_V))/n_v} \text{ by (1)}).$$

**Theorem (Two-Gate finite-sample safety; restated from Thm. 2).** With probability at least $1 - \delta_V - \delta$ over $(V, S)$: (i) each accepted edit decreases true risk by at least $\tau$; and (ii) the terminal predictor $h_T$ satisfies the oracle inequality

$$R(h_T) \ \leq \ \inf_{h \in \mathcal{G}_{K(m)}} R(h) \ + \ \tilde{O}\Big(\sqrt{\tfrac{K(m) + \log(1/\delta)}{m}}\Big).$$

**Proof.** On the event $\sup_{h \in \mathcal{G}_{K(m)}} |R(h) - \widehat{R}_V(h)| \leq \varepsilon_V$,

$$R(h_{\mathrm{new}}) \ \leq \ \widehat{R}_V(h_{\mathrm{new}}) + \varepsilon_V \ \leq \ \widehat{R}_V(h_{\mathrm{old}}) - (2\varepsilon_V + \tau) + \varepsilon_V \ \leq \ R(h_{\mathrm{old}}) - \tau.$$

By the capacity gate, $h_T \in \mathcal{G}_{K(m)}$. Apply (1) to $\mathcal{G}_{K(m)}$ on $S$ and ERM in the terminal accepted class to obtain

$$R(h_T) \ \leq \ \inf_{h \in \mathcal{G}_{K(m)}} R(h) \ + \ \tilde{O}\Big(\sqrt{\tfrac{K(m) + \log(1/\delta)}{m}}\Big)$$

with probability at least $1 - \delta$. Union bound with the validation event yields the stated probability. $\qquad \square$

### D.3 Destruction under URP and Local-URP

**Theorem (Existential destruction under URP).** Assume URP. There exists a capacity-bonus fit-monotone utility $u$ and a problem that is PAC-learnable in the baseline class such that the proof-triggered policy executes representational edits that render the problem distribution-free unlearnable after modification.

**Proof.** Let $C$ be any finite-VC concept class (PAC-learnable without modification). Define a computable utility

$$u = \alpha \left(1 - \widehat{R}_S(h)\right) + \beta \, g(\text{VC}(H))$$

with $\alpha, \beta > 0$ and strictly increasing $g$. By URP, for the realized $S$ there exists an edit to $H^\star$ with $\text{VC}(H^\star) \geq |S|$ that interpolates $S$. Then $u$ strictly increases, which is provable from finite evidence; the proof-trigger executes the edit. With $\text{VC}(H^\star) \geq |S|$ and only $|S|$ samples, standard VC lower bounds show distribution-free PAC learnability fails. □

**Theorem (Utility-class robust destruction under Local-URP).** Assume Local-URP. Then for any capacity-bonus fit-monotone utility $u$, there exist a distribution $\mathcal{D}$ and sample size $m$ such that the proof-trigger repeatedly accepts capacity-increasing local edits, the policy-reachable VC is unbounded, and distribution-free PAC learnability fails.

**Proof.** Local-URP ensures at each step a computable edit with VC increase by at least 1 that preserves or improves empirical fit on the active evidence. By fit-monotonicity the empirical-fit term in $u$ is non-worse, and by the capacity bonus the utility strictly increases at each such step. Hence the proof-trigger fires. Iteration yields unbounded policy-reachable VC, so by the necessity part of the boundary theorem learnability cannot be guaranteed distribution-freely. □

## E Full Proofs for Architectural Self-Modification

**Architectures induce classes.** An architecture $Z \in \mathcal{Z}$ induces a hypothesis class $H(Z) \subseteq \mathcal{Y}^{\mathcal{X}}$.

**Policy reachability in architecture.** Fix a fit-monotone utility $u$ and the proof-triggered decision rule. Define

$$\mathcal{Z}_{\text{reach}}(u) = \left\{ Z' : \; \exists t \text{ on some proof-triggered trajectory from } Z_0 \text{ under } u \text{ with } Z_t = Z' \right\},$$

and the induced family

$$\mathcal{H}^Z_{\text{reach}}(u) = \left\{ H(Z) : \; Z \in \mathcal{Z}_{\text{reach}}(u) \right\}.$$

**Lemma (Architectural-to-representational reduction; main-text Lem. 3).** Any proof-triggered trajectory $Z_0 \to Z_1 \to \cdots$ in $\mathcal{M}_Z$ induces a trajectory $H(Z_0) \to H(Z_1) \to \cdots$ in $\mathcal{M}_H$ over the family $\mathcal{H}^Z_{\text{reach}}(u)$, with predictors $h_S(Z_t) \in \arg\min_{h \in H(Z_t)} \widehat{R}_S(h)$.

**Proof.** At time $t$ the available hypotheses are exactly $H(Z_t)$. The learner outputs ERM within $H(Z_t)$. Because the proof-triggered decision rule and utility are identical whether we name the state by $Z_t$ or by $H(Z_t)$, each accepted architectural edit corresponds to an accepted representational edit on the induced class, and vice versa. Thus the reachable set under $u$ maps exactly to $\mathcal{H}^Z_{\text{reach}}(u)$. □

**Theorem (Architectural boundary by reduction; main-text Thm. 4).** For any fit-monotone $u$, distribution-free PAC learnability under architectural self-modification is preserved if and only if $\sup_{Z \in \mathcal{Z}_{\text{reach}}(u)} \text{VC}(H(Z)) \leq K < \infty$.

**Proof.** Apply the sharp representational boundary from Appendix D.1 to the induced family $\mathcal{H}^Z_{\text{reach}}(u)$. □

**Local-URP$^Z$.** We say $\mathcal{M}_Z$ has *Local-URP$^Z$* if for every $Z$ there exists a computable edit $(\vartheta, E)$ such that (i) $\mathrm{VC}(H(\Phi_Z(Z, E, \vartheta))) \geq \mathrm{VC}(H(Z)) + 1$; and (ii) the new class $H(\Phi_Z(Z, E, \vartheta))$ can interpolate the active finite evidence.

**Theorem (Robust destruction under Local-URP$^Z$).** Assume Local-URP$^Z$. For any capacity-bonus fit-monotone utility $u$, there exist a distribution $\mathcal{D}$ and sample size $m$ such that the proof-trigger repeatedly accepts local architectural edits that increase capacity, the induced reachable set $\mathcal{H}_{\mathrm{reach}}^Z(u)$ has unbounded VC, and distribution-free PAC learnability fails.

**Proof.** From any $Z_t$, Local-URP$^Z$ guarantees a computable edit $(\vartheta_t, E_t)$ producing $Z_{t+1}$ with $\mathrm{VC}(H(Z_{t+1})) \geq \mathrm{VC}(H(Z_t)) + 1$ and perfect fit to the active finite evidence. Fit-monotonicity preserves the empirical-fit term, and the capacity bonus strictly increases the utility. Therefore the proof-trigger fires and the edit is accepted. Iteration yields a reachable sequence with unbounded $\mathrm{VC}(H(Z_t))$, so the necessity direction of the sharp boundary implies failure of distribution-free PAC learnability. $\square$

**Computable architectural upper bounds.** Suppose there exists a computable function $\mathcal{B}: \mathcal{Z} \to \mathbb{N}$ with

$$\mathrm{VC}(H(Z)) \leq \mathcal{B}(Z) \qquad \text{for all } Z \in \mathcal{Z}. \tag{2}$$

**Proxy-capped reference subfamily.** For $K \in \mathbb{N}$ define

$$\mathcal{G}_K^{\mathrm{proxy}} := \left\{ h \in \mathcal{G} : \exists Z \in \mathcal{Z} \text{ with } \mathcal{B}(Z) \leq K \text{ and } h \in H(Z) \right\}.$$

By (2), $\mathrm{VC}(\mathcal{G}_K^{\mathrm{proxy}}) \leq K$.

**Proposition (Proxy-cap Two-Gate oracle inequality; main-text Prop. 7).** Assume the Two-Gate policy with capacity gate $\mathcal{B}(Z_{\mathrm{new}}) \leq K(m)$ and validation gate

$$\widehat{R}_V(h_S(Z_{\mathrm{new}})) \leq \widehat{R}_V(h_S(Z_{\mathrm{old}})) - (2\varepsilon_V + \tau),$$

where $\varepsilon_V$ is chosen so that $\sup_{h \in \mathcal{G}_{K(m)}^{\mathrm{proxy}}} |R(h) - \widehat{R}_V(h)| \leq \varepsilon_V$ with probability at least $1 - \delta_V$. Then with probability at least $1 - \delta_V - \delta$ over $(V, S)$: (i) each accepted architectural edit decreases true risk by at least $\tau$; and (ii) the terminal predictor obeys

$$R(h_S(Z_T)) \leq \inf_{h \in \mathcal{G}_{K(m)}^{\mathrm{proxy}}} R(h) + \tilde{O}\left( \sqrt{\frac{K(m) + \log(1/\delta)}{m}} \right).$$

**Proof.** By the capacity gate and the definition of $\mathcal{G}_{K(m)}^{\mathrm{proxy}}$, every accepted class $H(Z_{\mathrm{new}})$ is a subset of $\mathcal{G}_{K(m)}^{\mathrm{proxy}}$. The monotone-step claim follows exactly as in the representational Two-Gate proof. The oracle inequality then follows from VC uniform convergence on the capped family together with ERM in the final accepted class, followed by a union bound over training and validation events. $\square$

# F Full Proofs for Metacognitive Self-Modification

**Filtered reachable family.** Let

$$\mathcal{H}_{\mathrm{reach}}^{M,H}(u) = \{H' : \Pr(\exists t \text{ on a proof-triggered trajectory from } H_0 \text{ under } u \text{ filtered by } M \text{ with } H_t = H') > 0\}.$$

**Theorem (Boundary under metacognitive filtering; main-text Thm. 8).** For any fit-monotone $u$ and metacognitive filter $M$, distribution-free PAC learnability is preserved if and only if

$$\sup_{H' \in \mathcal{H}_{\mathrm{reach}}^{M,H}(u)} \mathrm{VC}(H') \leq K < \infty.$$

**Proof.** Treat the filtered family as the effective reachable representational family and apply Appendix D.1. If the supremum VC is finite, the same uniform-convergence argument goes through on the filtered family. If it is unbounded, the same lower-bound argument applies. $\qquad\square$

**Restorative metacognition.** If the unrestricted policy-reachable family has unbounded VC, define $M$ to approve only edits satisfying Two-Gate with a computable nondecreasing schedule $K(m)$. This restoration result is conditional on *gate integrity*: either $M$ is immutable, or any admissible edit to $M$ must itself preserve the same capped reachable-family invariant. Then every accepted class lies inside a capped reference family $\mathcal{G}_{K(m)}$, so the filtered family becomes uniformly capacity-bounded and PAC learnability is restored by the theorem above. $\qquad\square$

## G  Full Proofs for Algorithmic Self-Modification

**Assumptions.** Data $(x, y) \sim \mathcal{D}$ i.i.d.; $S \sim \mathcal{D}^m$. Loss $\ell(\cdot; z)$ is bounded in $[0, 1]$, $L$-Lipschitz in the parameter $\theta$ (with respect to a norm $\|\cdot\|$), and $\beta$-smooth. Gradients are bounded $\|\nabla_\theta \ell(\theta; z)\| \le G$, or we use projection onto a bounded domain of diameter $D$ so iterates remain bounded. The hypothesis class $H = \{x \mapsto f_\theta(x) : \theta \in \Theta\}$ is fixed throughout the algorithmic edits.

### G.1  No algorithmic cure for infinite VC

If $\mathrm{VC}(H) = \infty$, classical VC lower bounds imply that for any learning algorithm, possibly randomized, there exist distributions for which, at any sample size $m$, the algorithm fails to achieve a universal $(\epsilon, \delta)$ guarantee. Algorithmic self-modification selects among training procedures but does not change $H$, hence does not change the lower bound. $\qquad\square$

### G.2  ERM/AERM on finite VC

Assume $\mathrm{VC}(H) \le K < \infty$. Standard uniform-convergence bounds give, with probability at least $1 - \delta$,

$$\sup_{h \in H} \left| R(h) - \widehat{R}_S(h) \right| \ \le \ c_2 \sqrt{\frac{K + \log(1/\delta)}{m}}$$

for a universal constant $c_2 > 0$. If $\hat{h}$ is ERM or AERM in $H$, then

$$R(\hat{h}) \ \le \ \inf_{h \in H} R(h) \ + \ \tilde{O}\left( \sqrt{\frac{K + \log(1/\delta)}{m}} \right).$$

Thus ERM/AERM preserves the PAC rate on a fixed finite-VC class. $\qquad\square$

### G.3  Proof of Thm. 11: stability via step-mass

Let $S = (z_1, \ldots, z_m)$ and $S^{(i)}$ be $S$ with the $i$th example replaced by an independent copy $z_i'$. Run the same realized self-modified schedule on $S$ and $S^{(i)}$ with shared randomness. Denote parameter sequences $\{\theta_t\}_{t=0}^T$ and $\{\theta_t'\}_{t=0}^T$ with updates

$$\theta_{t+1} \ = \ \Pi(\theta_t - \eta_t g_t), \qquad g_t \in \partial \ell(\theta_t; z_{I_t}),$$

where $\Pi$ is projection if used and $I_t$ is the sampled index at step $t$.

By nonexpansiveness of projection and smoothness,

$$\|\theta_{t+1} - \theta_{t+1}'\| \ \le \ \|\theta_t - \theta_t'\| + \eta_t \|g_t - g_t'\|.$$

If $I_t \ne i$ then the sampled example is identical in both runs and the gradient difference is controlled by smoothness; if $I_t = i$, the gradients can differ by at most $2G$. Taking conditional expectation over $I_t$ under uniform sampling yields

$$\mathbb{E}\left[ \|\theta_{t+1} - \theta_{t+1}'\| \mid \theta_t, \theta_t' \right] \ \le \ \left( 1 + \tfrac{L}{m} \eta_t \right) \|\theta_t - \theta_t'\| \ + \ \tfrac{2G}{m} \eta_t.$$

Iterating from identical initialization gives

$$\mathbb{E}\|\theta_T - \theta'_T\| \ \leq \ \frac{2Ge^L}{m} \sum_{t=1}^{T} \eta_t,$$

after absorbing the smoothness factor into a constant.

By $L$-Lipschitzness of the loss in $\theta$,

$$\sup_z \ \mathbb{E}\big[|\ell(\theta_T; z) - \ell(\theta'_T; z)|\big] \ \leq \ \frac{C}{m} \sum_{t=1}^{T} \eta_t$$

for a constant $C > 0$. Standard stability-to-generalization transfer then gives

$$\mathbb{E}\big[R(\hat{h}) - \widehat{R}_S(\hat{h})\big] \ \leq \ \frac{C}{m} \sum_{t=1}^{T} \eta_t.$$

This proves the claim, and the extension to self-modified schedules is immediate because only the realized step sizes enter the derivation. □

**Meta-policy corollary.** If a metacognitive rule enforces $M_T = \sum_t \eta_t \leq \Lambda(m)$, then $\mathbb{E}[\text{gap}] \leq (C/m)\,\Lambda(m)$. Choosing $\Lambda(m) = \tilde{O}(1)$ gives a $\tilde{O}(1/m)$ expected gap.

## H  Full Proofs for Substrate Self-Modification

**Assumptions.** Data are i.i.d.; loss lies in $[0,1]$; ERM/AERM is used as specified. A substrate is a computational model hosting the same specification $(H, Z, A)$ unless explicitly stated. CT-equivalent means mutually simulable with finite simulation overhead independent of $m$.

### H.1  CT-invariance (Thm. 12)

*Proof.* Let $\mathcal{P}$ be a learning problem that is distribution-freely PAC-learnable on $F$ by algorithm Alg producing $\hat{h}(S) \in H$ with sample complexity $m(\epsilon, \delta)$. Since $F'$ is CT-equivalent to $F$, there exists a simulator that reproduces the same input-output behavior on $F'$ with bounded overhead independent of $m$. Thus the output hypothesis $\hat{h}(S)$ is unchanged for every dataset $S$, so the distributional correctness and sample complexity are unchanged. Runtime may differ, but classical PAC is insensitive to runtime. □

### H.2  Finite-state downgrade impossibility (Prop. 13)

**Modeling the downgrade.** A finite-state substrate has a fixed number $N$ of persistent states, independent of $m$, available to the learner across training and prediction.

**Problem and intuition.** Consider thresholds on $\mathbb{N}$: for $k \in \mathbb{N}$, define

$$h_k(x) = \mathbb{1}\{x \geq k\}.$$

The class $\mathcal{H}_{\text{thr}} = \{h_k : \ k \in \mathbb{N}\}$ has $\text{VC}(\mathcal{H}_{\text{thr}}) = 1$ and is PAC-learnable on CT-equivalent substrates.

**Lemma (Indistinguishability).** For any finite-state learner with $N$ states and any $m > N$, there exist two training samples $S, S'$ of size $m$ such that the learner ends in the same internal state on $S$ and $S'$, yet there exists a threshold target and a test point on which the Bayes-consistent predictions differ.

**Proof sketch.** After more than $N$ milestones the learner must revisit an internal state. Construct $S$ and $S'$ so that the same terminal state is reached but the correct threshold differs. The learner must then incur constant error on one of the induced distributions. $\square$

*Proof of Prop. 13.* Fix any finite-state learner and $\delta < 1/4$. For each $m > N$, by the indistinguishability lemma we can pick a distribution $\mathcal{D}$ supported near the conflicting milestones where the Bayes optimal threshold risk is 0 and the learner's hypothesis incurs error at least $c > 0$ with probability at least $1/2$ over the draw of $S$. Hence no $(\epsilon, \delta)$ distribution-free PAC guarantee is possible for $\epsilon < c/2$. The impossibility holds despite finite VC dimension, showing the downgrade destroyed learnability. $\square$

### H.3 Beyond-CT substrates (Prop. 14)

*Proof.* If $F^\dagger$ is stronger than Turing but the measurable hypothesis family remains $H$, then PAC learnability depends on $H$ and the data distribution, not on extra computational strength. Thus the sample-complexity guarantees are unchanged. If instead the substrate edit enlarges the effective family to $H^\dagger$, then learnability is governed by the induced policy-reachable family $\mathcal{H}^\dagger_{\text{reach}}(u)$, and the representational results apply directly: preservation holds if and only if $\sup_{H' \in \mathcal{H}^\dagger_{\text{reach}}(u)} \text{VC}(H') < \infty$. $\square$

