# OpenReview forum: "On the Statistical Limits of Self-Improving Agents"
_TMLR — Decision pending for TMLR_

### Review · Reviewer_yfFL · 2026-04-30

**Summary Of Contributions:**

The paper studies PAC-style generalization in a sequential setting that mimics current “agentic” approaches. Given a labeled classification dataset split into training and validation sets and some initial hypothesis class, the agent fits a model from said class to the training set, checks its validation accuracy, and reacts by modifying some facet of its system, e.g., the hypothesis class by increasing the depth of an MLP, the optimizer, etc. The agent can also make meta decisions, such as modifying the criteria based on which these “edits” are accepted. For instance, the amount of improvement in validation accuracy that is sufficient for an edit to be deemed worthwhile.

It is easy to see that some of these edits can make the ultimate classifier useless from a generalization perspective. On the one hand, increasing the capacity of the hypothesis class unboundedly can result in overfitting. On the other hand, restricting the system excessively can result in underfitting.

The authors propose five axes along which these edits can be classified: algorithmic (e.g., optimizer change, learning rate change, stopping rules, etc.); representational (e.g., kernel feature expansion); architectural (e.g., for deep nets, increasing depth or modifying activation functions, etc.); metacognitive (i.e., modifying the agent’s own decision rules that govern the sequence of models it constructs); substrate (i.e., modifying the computation model itself).

For each of these, a section analyzes the result of edits of that type assuming the other four are fixed. Representational modification seems to be the most important axis to consider, as most of the other axes can reduce to it. The authors propose what they call a “two-gate guardrail” to safeguard against edits that dangerously increase model capacity: accept an edit iff the validation accuracy increases by a large enough amount AND the VC dimension of the new model is not larger than would be appropriate for the amount of training/validation data available to the agent. This serves as the key takeaway of the paper, as argued in the Outlook and Conclusion sections.

**Additional Comments:**

I am no learning theory expert and thus could not verify the proofs for correctness.

**Audience:**

Yes

**Audience Explanation:**

This is a conservative “Yes” at this point. I felt somewhat interested in what the paper had to offer. However, in the problem setting studied here, the agent’s decision rule seems to be super restricted in that it cannot increase model capacity too much at risk of overfitting, making the conclusions of the paper somewhat obvious. I don’t know if theoreticians will find the work particularly insightful.

**Broader Impact Concerns:**

None.

**Claims And Evidence:**

Yes

**Claims Explanation:**

I did not verify the proofs in the appendix in detail as they are outside my expertise. At quick glance, the claims are reasonable and the proofs seem to utilize basic techniques from generalization theory bounds.

**Requested Changes:**

Major:
- Hyperparameter tuning is mentioned briefly in a single sentence of section 2.2. However, given the problem setup, it seems to be far more related to the paper. Most hyperparameter tuning is sequential, uses a train/validation split, can increase/decrease capacity, and thus involves the same tradeoffs of the self-modification problem herein. Surely there is some learning theory for hyperparameter tuning which applies here too?

- The architectural axis seems to be specific to deep neural networks. Why not fold it into Representational? I am not sure both are needed. You do end up reducing the former to the latter anyways.

- High-level motivation: it would seem to me that there is a substantial mismatch between the notion of a self-improving agent used in the paper vs. how such agents are currently used in practice. It would seem to me that current agent applications are ones where overfitting is in some sense desirable, i.e., you want the sequence of agent decisions to arrive at a piece of code or the like that passes certain tests. I do however think your framework bears resemblance to this very recent trend: https://github.com/karpathy/autoresearch. Perhaps grounding the paper in a problem setting that is more empirically justified will strengthen the overall story and results.

Minor:
- Some references should be parenthesized, e.g., all references in the last paragraph “Evidence for this…” of Page 1. Similarly, the modification axes A/H/Z/F/M are introduced top of page 4, way before they are defined meaningfully.
- Section 3: the utility function is invoked before t, s_t, E_t, or {Env}_t are defined.
- Some of the writing is redundant, e.g., in page 6, “Setting, fixed versus modifiable”, the last sentence repeats earlier assumptions/notation.

---

> ### Author Response · Authors · 2026-06-09
>
> We thank the reviewer for the careful reading and constructive suggestions. We will revise the paper to address each concern directly.
>
> First, we will add a clearer discussion of hyperparameter tuning and model selection, explaining that they are important special cases of our framework. The broader self-modification setting differs because the agent may also modify the representation, architecture, update rule, acceptance criterion, and future edit process itself.
>
> Second, we will clarify why the architectural axis is kept separate from representation. We agree that, statistically, architectural edits reduce to the induced hypothesis family, but we keep the axis separate because architecture-level changes are operationally distinct in modern systems.
>
> Third, we will strengthen the motivation with a more concrete running example based on modern code/research agents. In these settings, passing visible tests may be locally useful, but the same utility-learning tension appears when edits improve observed performance while weakening generalization to hidden tests, future tasks, or deployment settings.
>
> We will also address the minor writing and notation issues, including reference formatting, earlier definition of the five axes, earlier definition of the utility-function notation, and removal of redundant phrasing.
>
> Overall, these are directly addressable revisions that improve the framing and presentation while leaving the core content and proofs intact. We believe the paper’s main contribution remains meaningful: it gives TMLR readers a useful framework for understanding when self-modification preserves reliable learning and when it undermines it. We respectfully ask the reviewer to support acceptance of the revised manuscript.

---

> > ### Comment · Reviewer_yfFL · 2026-06-10
> > **Revision?**
> >
> > Thank you for the response! Since there are many suggested changes, it would help to see a revised version of the paper before I submit a final recommendation. Do you have one in the works?

---

> > > ### Author Response · Authors · 2026-06-14
> > > **Revision**
> > >
> > > Thank you for the acknowledgement. We have attached our latest revised version, where we have made the requested changes. Please let us know if there are any additional questions or concerns!

---

> > > > ### Comment · Reviewer_yfFL · 2026-06-23
> > > > **List of changes**
> > > >
> > > > thank you. I see some of the changes but comparing versions side-by-side is tricky. A list of changes would be very helpful.

---

> > > > > ### Author Response · Authors · 2026-06-24
> > > > > **List of changes**
> > > > >
> > > > > Below is a list of revisions.
> > > > >
> > > > > **1. Expanded the practical motivation in the “Introduction” section.**
> > > > >
> > > > > We rewrote the opening around coding agents that edit programs until tests pass, AutoML loops that select models and hyperparameters from validation feedback, and code/tool-use systems. It also explains that an agent can improve on visible tests yet overfit its modification sequence and fail on hidden tests, future tasks, or deployment settings. This motivates the utility-learning tension.
> > > > >
> > > > > **2. Added a concrete running example in the “A Simple Multi-Axis Example” section.**
> > > > >
> > > > > We added an example separating update-rule selection from enlarging a polynomial feature family. It shows that optimizer switching changes search within a fixed family, while representation growth changes what the learner can express; a global capacity cap preserves learnability, whereas repeated locally beneficial expansions can outgrow the available evidence.
> > > > >
> > > > > **3. Expanded hyperparameter tuning and model selection in the “Modern mechanisms for self-improvement” subsection of “Related Work” and the “Hyperparameter tuning” paragraph of “Algorithmic Self-Modification.”**
> > > > >
> > > > > The revision distinguishes **capacity-neutral choices**—optimizer, learning-rate, batch schedule, and stopping rule—from choices that **change the effective hypothesis family**, such as depth, width, kernel features, regularization, retrieval size, and number of experts. We clarify that standard validation search applies when the candidate family is fixed ex ante; when an agent can change that set itself, validation gains alone are insufficient unless the policy-reachable family remains controlled.
> > > > >
> > > > > **4. Clarified and formally reduced the architectural axis in the “Architectural Self-Modification” section.**
> > > > >
> > > > > We now state that architecture can be folded into representation from a PAC perspective, since it matters through the family it induces, but retain it as a distinct operational intervention point. Its revised definition covers topology and information flow—not only deep networks—including routing, planners, tool interfaces, memory, depth, width, and expert mixtures. We also added the formal “Architectural to representational reduction,” showing that the relevant object is the induced reachable hypothesis family.
> > > > >
> > > > > **5. Clarified the relationship to standard online and continual learning in the “Introduction” and the “Relation to standard online and continual learning” paragraph of “Algorithmic Self-Modification.”**
> > > > >
> > > > > We clarify that update-rule switching is not a replacement for online or continual learning. It is a structured policy-selection problem within those settings: the learner remains history-dependent, but finite-evidence update selection can overfit when the reachable update-rule family is too flexible for the available evidence.
> > > > >
> > > > > **6. Made the technical novelty explicit in the “Technical novelty” paragraph of the “Introduction” section.**
> > > > >
> > > > > We added a dedicated paragraph clarifying that the contribution is not a new uniform-convergence inequality. Rather, it is the **policy-reachable family**: the effective family induced by the agent’s modification policy, including representation, architecture, update dynamics, candidate-set, and approval-rule changes. We distinguish it from the initial class, an unconstrained super-class, and a single post-edit model.
> > > > >
> > > > > **7. Expanded and qualified assumptions in the “Scope of the claims” paragraph of the “Introduction” and the “Limitations and Scope” section.**
> > > > >
> > > > > We now state that the results are a baseline under distribution-free PAC assumptions: fixed-distribution i.i.d. data, bounded loss, independent validation data where used, and one-axis-at-a-time analysis unless a joint reachable family is defined. The expanded limitations identify requirements for adaptive, nonstationary, or interactive extensions, including concentration tools, drift or mixing assumptions, reusable holdouts, and online-to-batch or covariate-shift arguments.
> > > > >
> > > > > **8. Reorganized definitions, notation, and presentation in the “Setup and Five-Axis Decomposition” section and the “Introduction.”**
> > > > >
> > > > > The five axes and complete learner state are now defined before use. We clarify that they concern, respectively, update dynamics; the expressible family; topology and information flow; computation and memory semantics; and modification scheduling and acceptance.
> > > > >
> > > > > We also moved the definitions of time, learner state, finite evidence, and environment context before utility; distinguished utility, population risk, and empirical risk; and defined the utility-learning tension precisely. Finally, we corrected citation formatting, consolidated repeated fixed-versus-modifiable assumptions, and removed redundant phrasing.
> > > > >
> > > > > The core theorem and proof strategy are unchanged, while the revision now more directly connects the framework to hyperparameter tuning, online learning, modern code agents, architectural changes, and the limits of the i.i.d. analysis.

---

### Review · Reviewer_uFQ9 · 2026-05-13

**Summary Of Contributions:**

This paper proposes a PAC-style framework for self-modifying agents. It decomposes self-modification into five axes and argues that distribution-free learnability is preserved iff the policy-reachable hypothesis family has uniformly bounded capacity. It also proposes a Two-Gate guardrail based on validation improvement and a capacity cap.

**Audience:**

Yes

**Audience Explanation:**

Some TMLR readers interested in learning theory, agentic systems, and AI safety may find the reachable-family perspective and five-axis decomposition useful. However, the audience may be somewhat limited, since the results are mainly a PAC/VC reframing and do not yet provide strong practical guidance for modern self-improving agents.

**Broader Impact Concerns:**

I do not see major immediate broader impact concerns. The paper is primarily theoretical and aims to characterize when self-modification preserves learnability.

**Claims And Evidence:**

No

**Claims Explanation:**

The main claims are supported under the paper’s stated PAC/VC assumptions, and the arguments are mostly clear. However, the evidence is mainly theoretical and follows closely from classical uniform convergence once the reachable hypothesis family is defined. I am not fully convinced that this supports the broader claims about self-improving agents in realistic settings, especially under adaptive data collection, non-i.i.d. interaction, or RL-style feedback.

**Requested Changes:**

The authors should clarify the main technical novelty beyond applying classical VC/PAC theory to the policy-reachable hypothesis family. They should also narrow or qualify the broader claims about self-improving agents, since the current results rely on restrictive i.i.d. assumptions.

I also encourage the authors to provide more concrete examples of non-vacuous capacity proxies for modern LLM-based agents, and to discuss how the framework would change under adaptive data collection, non-stationarity, or RL-style feedback.

The paper would also benefit from a stronger running example showing how the five axes lead to different reachable families and when the proposed guardrail gives nontrivial insight.

---

> ### Author Response · Authors · 2026-06-09
>
> We thank the reviewer for the careful feedback. We agree that the original version should be clearer about both the scope of the claims and the technical novelty of the paper.
>
> **Relevance to modern agents and scope of claims.**
> We agree that modern self-improving agents often operate under adaptive data collection, non-i.i.d. interaction, or RL-style feedback. Our current results do not claim to solve those full settings. We will revise the paper to state more clearly that our formal theorem is a PAC-style baseline under standard i.i.d. assumptions.
>
> At the same time, we believe the framework remains fundamentally relevant to modern agents. Even when the data are adaptive or nonstationary, a self-improving agent still induces a reachable family of policies through the edits it is allowed to make. The central question remains whether that reachable family is structurally controlled enough to support reliable generalization.
>
> Many current agents improve by trying edits to prompts, tools, architectures, search procedures, or acceptance criteria, then keeping edits that improve some validation signal. Our point is that local improvement alone is not enough: if these edits expand the reachable family without structural control, the agent can lose the statistical conditions needed for reliable generalization. This remains an important concern even when future work extends the analysis beyond i.i.d. data.
>
> **Technical novelty beyond classical VC/PAC theory.**
> We agree that the proof techniques are closely related to classical uniform convergence. The novelty is not a new VC inequality by itself. The contribution is identifying the policy-reachable hypothesis family as the right object of analysis for self-modifying agents.
>
> Classical learning theory usually assumes the hypothesis class, update rule, and selection protocol are fixed externally. In our setting, the agent can modify these objects itself. The main result shows that different kinds of self-modification—representational, architectural, algorithmic, metacognitive, and substrate-level—share the same statistical bottleneck: whether the induced reachable family remains uniformly capacity-bounded.
>
> **Examples and practical grounding.**
> We will add a stronger running example based on a modern LLM/code agent. For example, such an agent may edit its prompt, add tools, change its search procedure, modify its acceptance threshold, or move to a larger model class. Each of these edits changes the family of policies the agent can reach. The Two-Gate guardrail is useful because it does not simply ask whether the edit improves validation performance; it also asks whether the resulting reachable family remains within a capacity regime supported by the available data.
>
> We will also add concrete capacity proxies for modern agents, such as finite edit libraries, bounded tool/action spaces, bounded prompt or program length, bounded search depth, caps on the number of accepted self-edits, hidden validation tests, and architecture/model-family caps.
>
> **Summary and request.**
> We believe these revisions address the reviewer’s concerns by narrowing the paper’s claims while clarifying its main contribution. The paper provides a useful framework for understanding when self-improvement preserves reliable learning rather than undermining it. We respectfully ask the reviewer to consider supporting acceptance of the revised manuscript.

---

### Review · Reviewer_yDHF · 2026-05-26

**Summary Of Contributions:**

The paper introduces a theoretical framework for the analysis of self-improving learning agents, centred on a five-axis decomposition (representational, architectural, meta-cognitive, algorithmic, and substrate) with a separate decision layer that disentangles incentives from learning behaviour. The central contribution is the identification of a sharp *utility-learning tension*: utility-driven self-modifications that improve immediate performance can simultaneously erode the statistical pre-conditions for reliable learning and generalization. The analysis shows that for such agents, PAC guarantees can be preserved **if and only if** the family of reachable policies (induced by self-modification) has uniformly bounded capacity. This bi-conditional sharpness is a key strength: unbounded capacity does not merely risk PAC failure, it necessarily causes it. Two guardrails (a validation margin and a capacity cap) are introduced to keep agents on the safe side of this boundary.

**Novelty**
1. **Theoretical framework for self-improving agents.** Developed a learning-theoretic framework for agents that self-modify not just weights but their update rules, architecture, computational substrate, and meta-reasoning. These domains that classical learning theory assumed to be fixed.
2. **Sharp iff learnability boundary.** Proved that distribution-free PAC learnability is preserved under self-modification if and only if the policy-reachable family has uniformly bounded capacity (measured by VC dimension or an equivalent uniform-convergence notion).
3. **Axis reductions.** Architectural and metacognitive edits reduce to induced hypothesis families, and substrate changes matter only via computability or the induced family. This means the learnability boundary depends solely on the supremum capacity of the reachable family, a unifying result across all five axes.
4. **Two-Gate guardrail.** A computable accept/reject rule (validation improvement by margin $tau$ plus a capacity cap $K(m)$) that ensures monotone true-risk progress and yields a VC-rate oracle inequality for the final predictor, a concrete and practically actionable guarantee.


**Hypothesis and Claims**
**Claim 1.** Distribution-free PAC learnability is preserved under self-modification if and only if the policy-reachable family has uniformly bounded capacity under standard i.i.d. assumptions. The bi-conditional is sharp: when capacity can grow without limit, utility-rational self-modifications can render previously learnable tasks unlearnable.

**Claim 2.** The Two-Gate guardrail a validation margin $tau$ combined with a capacity cap $K(m)$ is sufficient to enforce the capacity boundary of Claim 1, ensuring monotone true-risk steps and finite-sample safety at VC rates.


**Questions**
1. **Page 12:** The paper suggests that the introduced framework provides a platform to analyse simultaneous modifications by several components via the induced reachable policy family. While this is true, can the framework also quantify the relative contributions of each axis to the induced policy? Furthermore, could the authors share any insight into which axis pushes the induced policy family the most out of bounds?

2. **i.i.d. assumption realism:** The paper's results rest on standard i.i.d. assumptions. Self-improving agents, however, are likely to operate in non-stationary environments where this assumption is routinely violated. Could the authors comment on how fragile the learnability boundary is under distribution shift, or sketch what additional conditions would be needed to extend the results beyond i.i.d.?



**Limitations and Assumptions**
The paper operates throughout under standard i.i.d. assumptions and analyses each of the five axes in isolation. Both are reasonable starting points but represent meaningful restrictions on the scope of the results. Real self-improving agents are likely to modify multiple axes simultaneously, and the paper acknowledges that the multi-axis case remains an open problem. Additionally, the paper does not address non-stationary or adversarial data settings, which are arguably the most realistic environments for agents that actively reshape their own learning mechanisms. Future work that relaxes these assumptions, or that quantifies the degradation of guarantees under mild distribution shift, would substantially strengthen the practical relevance of the framework.

**Audience:**

Yes

**Audience Explanation:**

The paper introduced a new analytical framework that advances theoretical studies of self-improvement learning methods via self-modification.

**Claims And Evidence:**

Yes

**Claims Explanation:**

The claims in the paper are supported by theoretical evidence and analysis.

**Requested Changes:**

1. Page 3: Define the terms $s_t$, $E_t$, and $\text{Env}_t$ where the expression $u(s_t, E_t, \text{Env}_t)$ is first introduced.
2. Page 4: Define the term $R(\hat{h})$ 4 at the first use.

---

> ### Author Response · Authors · 2026-06-09
>
> We thank the reviewer for the careful reading and for recognizing the main contributions of the paper, including the five-axis framework, the sharp reachable-family learnability boundary, and the Two-Gate guardrail. We appreciate the constructive questions and will revise the manuscript to make these points clearer.
>
> **Relative contributions of each axis.**
> The current framework measures the effect of self-modification through the induced reachable policy family. It does not yet give a full causal decomposition of how much each axis contributes when several axes are modified at the same time. We agree this is an important direction. In the revision, we will clarify that the single-axis analyses should be understood as controlled reductions: each axis is isolated to show how it changes the reachable family.
>
> For simultaneous modifications, one possible way to quantify an axis’s contribution is to compare the capacity of the reachable family with and without that axis enabled. However, these contributions may not be additive, since axes can interact. For example, a metacognitive edit may change the future acceptance rule, which can then allow larger representational or architectural changes.
>
> There is also no universal ranking of which axis is most likely to push the reachable family out of bounds. Representational and architectural edits are the most direct sources of capacity growth, because they expand the expressive family itself. Metacognitive edits may be especially dangerous indirectly, because they can change the rules that govern future edits. Algorithmic and substrate edits matter when they change the effective family of policies the agent can reach.
>
> **i.i.d. assumption and distribution shift.**
> We agree that the i.i.d. assumption is a meaningful restriction. The current result should be read as a clean baseline theorem: under standard i.i.d. PAC assumptions, learnability is preserved exactly when the policy-reachable family remains uniformly capacity-bounded.
>
> Under distribution shift, this condition is not sufficient by itself. If the training, validation, and deployment distributions differ, then capacity control must be paired with additional assumptions that relate these distributions. In the revision, we will add a clearer discussion of what would be needed beyond the i.i.d. setting, such as stability assumptions, mixing conditions, martingale-style tools for adaptive data collection, domain-adaptation assumptions for controlled shift, or online/regret-style guarantees for interactive settings.
>
> **Limitations and future work.**
> We agree with the reviewer that analyzing each axis in isolation and assuming i.i.d. data are reasonable starting points but important limitations. Real self-improving agents may modify multiple axes at once and may operate in non-stationary or adversarial environments. We will revise the limitations section to state this more directly and to frame multi-axis attribution and non-i.i.d. extensions as important future work, but still highlight the importance of this work as a foundational starting point.
>
> **Minor requested changes.**
> We will make the requested notation fixes in the final version.
>
> We believe these revisions clarify the paper’s main contribution and make its scope more precise. We hope the reviewer agrees that the revised manuscript offers TMLR readers a useful framework for understanding when self-improvement preserves reliable learning, and we respectfully ask them to support acceptance.

---

### Decision · Action_Editor_jdpf · 2026-07-05

**Recommendation:** Accept as is

**Additional Comments:**

From the reviews, emerges a picture of a paper that studies a very relevant problem in the AI community, with the recent success of LLM-powered agents, but with a set of assumptions (especially i.i.d. data) that may defeat the purpose of providing statistical ground to realistic agents implementations.

I believe the provided framework, despite the limitations coming from the strong assumptions, could be a useful stepping stone towards a more realistic study of the statistical properties of self-improving agents, and I thus provide an accept recommendation.

**Audience:**

Yes

**Audience Explanation:**

I think the paper can draw attention of a potentially broad community of TMLR, given the recent trends of agentic AI and auto-research, although the technical results are mostly directed to the statistical learning community.

**Claims And Evidence:**

Yes

**Claims Explanation:**

The reviewers agree that the claims of the paper are validated by the presented theory, although they cast some doubts on whether the considered assumptions would make the results meaningful for realistic settings.